# Worldwide Review and Meta-Analysis of Cohort Studies Measuring the Effect of Mammography Screening Programmes on Incidence-Based Breast Cancer Mortality

**DOI:** 10.3390/cancers12040976

**Published:** 2020-04-15

**Authors:** Amanda Dibden, Judith Offman, Stephen W. Duffy, Rhian Gabe

**Affiliations:** 1Centre for Cancer Prevention, Wolfson Institute of Preventive Medicine, Queen Mary University of London, Charterhouse Square, London EC1M 6BQ, UK; a.dibden@qmul.ac.uk (A.D.); r.gabe@qmul.ac.uk (R.G.); 2Comprehensive Cancer Centre, School of Cancer & Pharmaceutical Sciences, Faculty of Life Sciences & Medicine, King’s College London, Innovation Hub, Guys Cancer Centre, Guys Hospital, Great Maze Pond, London SE1 9RT, UK; judith.offman@kcl.ac.uk

**Keywords:** breast cancer, screening, mammography, incidence-based mortality

## Abstract

In 2012, the Euroscreen project published a review of incidence-based mortality evaluations of breast cancer screening programmes. In this paper, we update this review to October 2019 and expand its scope from Europe to worldwide. We carried out a systematic review of incidence-based mortality studies of breast cancer screening programmes, and a meta-analysis of the estimated effects of both invitation to screening and attendance at screening, with adjustment for self-selection bias, on incidence-based mortality from breast cancer. We found 27 valid studies. The results of the meta-analysis showed a significant 22% reduction in breast cancer mortality with invitation to screening, with a relative risk of 0.78 (95% CI 0.75–0.82), and a significant 33% reduction with actual attendance at screening (RR 0.67, 95% CI 0.61–0.75). Breast cancer screening in the routine healthcare setting continues to confer a substantial reduction in mortality from breast cancer.

## 1. Introduction

Reviews of randomised controlled trials (RCTs) of mammography screening estimate that invitation to screening reduces risk of death from breast cancer by around 20% [1,2]. However, as the majority of RCTs were carried out over 30 years ago, they do not take account of changes in breast cancer incidence [3], mortality [4], screening techniques and treatments that have occurred over time. Furthermore, the results may not be representative of the effectiveness of individual population mammography screening programmes [5], which are affected by factors such as varying round lengths, radiographer skill and technology [6]. While RCTs provide reliable evidence and proof of principle that mammography screening is likely to be beneficial, once population screening programmes have been introduced, randomisation to a non-interventional control is no longer ethical and it is necessary to measure the effectiveness of screening in practice through observational studies.

Cohort studies have been used to achieve this objective but there can be important, subtle differences in methods of analysis used. One method is to use incidence-based mortality (IBM) [7], where deaths from breast cancer are only included in women diagnosed after screening has been introduced [8]. This avoids contamination of deaths in the screening period of women who were diagnosed prior to the start of screening, which would bias results against screening [9]. The aim of this review is to provide an overview of all IBM studies evaluating the impact of mammography screening on breast cancer mortality and to establish an up to date estimate of the long term benefit of breast screening.

## 2. Materials and Methods

### 2.1. Search Strategy

A systematic search of PubMed was performed in October 2019 with search terms based on those used by Njor et al. in their review of European IBM studies (Euroscreen project) [8]. Inclusion criteria were that (i) the study used IBM in the analysis, (ii) the study outcome was breast cancer mortality and (iii) the paper was in English. No restrictions were placed on age of study participants included to enable the inclusion of as many studies, and hence women, as possible.

### 2.2. Selection of Sources of Evidence

The titles and abstracts were initially assessed for relevance. A random selection of 100 papers were independently assessed by three reviewers (A.D., S.W.D. and J.O.) for accuracy. Following observation of more than 90% agreement among reviewers, the remainder of the papers were assessed by one reviewer (A.D.). The main text of the potentially eligible papers was then assessed by two reviewers (A.D. and S.W.D.) in order to make a final decision regarding inclusion in the review. We prepared a list of variables to extract from each paper (if available). These included programme characteristics, person years accrued and relative risks associated with invitation and/or exposure to screening as well as the proportion participating in screening (the latter to assist in correction for self-selection bias).

### 2.3. Statistical Methods

Random effects meta-analyses were undertaken to obtain overall estimates of the effects of (i) invitation to screening and (ii) attendance to screening on risk of breast cancer mortality [10]. It is important to note that when assessing the effect of invitation on mortality, it pertains to populations offered screening and the effect of attendance pertains to women who actually take up the offer of screening and is thus effected by the participation rate. Analyses were repeated stratified by age group (i) 50 and over, (ii) under 50. We chose age 50 to stratify the data as the majority of studies reported on the effects of screening in women aged 50–69, reflecting many national screening programmes, and in order to provide separate evidence in women under 50 years where possible as there has been uncertainty about whether screening younger women is effective and hence, cost-effective. Heterogeneity between studies was assessed using the χ^2^ test. Where studies used overlapping data, the largest study was chosen on the basis of better precision with a smaller variance. 

Statistical analyses were conducted using Stata Version 13 (StataCorp, College Station, TX, USA) [11].

#### Adjustment for Self-Selection Bias

Studies have shown that women who do not comply with an invitation to screening usually, but not invariably, have a higher risk of breast cancer mortality than those who choose to attend, resulting in a bias in favour of screening [12]. In order to account for such self-selection bias in studies reporting the effect of attending, we used the statistical adjustment proposed by Duffy et al. [13]. This uses the relative risk of attenders versus non-attenders from the current study, the participation rate, and the risk of death in non-attenders versus uninvited from an appropriate external source (Table 1). The relative risk of non-attenders versus uninvited women of 1.17 (95% CI 1.08–1.26) reported in the Swedish Organised Service Screening Evaluation Group (SOSSEG) study was used in this review as it was a large population based service screening study investigating IBM [14].

## 3. Results

### 3.1. Literature Selection

The literature search identified a total of 5232 titles from three searches performed in PubMed (see Appendix A for details of searches 1, 2 and 3), and 43 were deemed relevant for our review after assessment of abstracts and full text (Figure 1). Of these, four studies assessed the effectiveness of screening programmes outside Europe, one each from Canada [15] and the USA [16], and two from New Zealand [17,18]. The remaining 39 studies were European with twelve from Sweden [14,19,20,21,22,23,24,25,26,27,28,29], nine from Finland [30,31,32,33,34,35,36,37,38], five from Norway [39,40,41,42,43], four from both Italy [44,45,46,47] and Denmark [48,49,50,51], two from the Netherlands [52,53] and the UK [54,55] and one from Spain [56].

There was overlap between some papers, whereby authors used the same data to estimate the effect of different outcomes or updated results with longer follow-up. This resulted in sixteen exclusions (one paper from Denmark [51], four from Finland [31,34,35,37], two from Italy [44,46], three from Norway [39,42,43] and six from Sweden [19,21,22,26,27,29]). The remaining twenty-seven papers included in this review, representing independent populations, are summarized in Table 2.

### 3.2. Study Findings

Whilst the majority of studies included women in the age range 50–64 years, the youngest age of invitation was 35 years and the oldest 83 years. Most countries invite women every two years, with the range between 18 months and three years. Table 3 shows the unadjusted relative risk for the effect of invitation and attendance on the outcome of incidence-based breast cancer mortality as reported in the studies, and corresponding relative risks adjusted for self-selection bias as described above. The effect sizes and the participation rates reported in the studies suggest differences in risk of breast cancer mortality within countries as well as between countries. Participation rates ranged from 44% in Canada to above 90% in Finland.

The studies reviewed used one or more of three types of comparison groups used to estimate breast cancer mortality in an uninvited population: contemporaneous, regional and historical. (i) The contemporaneous comparison group compared women not yet invited, during the same time period and in the same region, as the women invited. (ii) The regional comparison group is often concurrent to the invited women, but in a region not yet invited. (iii) The historical comparison group compares women invited with women from an epoch not yet invited.

Ten studies compared the screening population with a contemporaneous comparison group, five of which estimated the effect of invitation to screening and seven the effect of attending screening. The effect of invitation fell within a narrow range from 0.72 (95% CI 0.64–0.79) [41] to 0.81 (95% CI 0.64–1.01) [47] and the effect of attendance was 0.38 (95% CI 0.30–0.49) [17] to 0.67 (95% CI 0.49–0.97) [38].

A further six studies used a historical comparison group to compare the impact of introducing screening in a particular region or country, the majority of which reported both the effect of invitation and attendance. The range of the respective effect sizes was wider for invitation than the studies that used a contemporaneous comparison group but narrower for attendance at 0.58 (95% CI 0.44–0.75) [56] to 0.83 (95% CI 0.73–0.95) [18] and 0.52 (95% CI 0.46–0.59) [23] to 0.66 (95% CI 0.58–0.75) [32] respectively.

Only four studies used a regional comparison group without any adjustment for differences in underlying cancer incidence between regions. All studies estimated the effect of invitation, with the results ranging from 0.73 (95% CI 0.63–0.84) [55] to 0.94 (95% CI 0.68–1.29) [53]. Just one study reported the effect of attending screening and found a 29% reduction in breast cancer mortality (95% CI 0.62–0.80) [20] in women who attended screening compared to those who did not.

The remaining seven studies used a combination of regional and historical data, and again, all studies estimated the effect of invitation, with effect estimates ranging between 0.75 (95% CI 0.63–0.89) [50] to 0.97 (95% CI 0.73–1.28) [25], which is almost identical to the results of those studies that used a regional control group alone. The two studies that estimated the effect of attendance reported relative risks of 0.60 (95% CI 0.49–0.74) [50] and 0.68 (95% CI 0.59–0.79) [49] respectively.

### 3.3. Meta-Analysis by Age-Group of Women

There were twenty-two studies that assessed the effect of invitation to screening (Table 3 and Figure 2). All studies had a relative risk of less than or equal to 1, with the largest studies achieving statistical significance. The largest studies were those by SOSSEG [14] and Johns et al. [54] who found a 20–30% reduction in breast cancer mortality. The pooled rate ratio was 0.78 (95% CI 0.75–0.82) with significant heterogeneity (*p* < 0.001).

Fourteen studies reported on the effect of being screened, eight of which also reported on the effect of being invited. All but one study reported a statistically significant result with the largest studies again by SOSSEG [14] and Johns et al. [54] with relative risks of 0.54 and 0.55 respectively. This therefore led to a pooled estimate of 0.54 (95% CI 0.49–0.59) with significant heterogeneity at *p* < 0.001. However, as discussed previously, the effect estimate of being screened is likely to be subject to self-selection bias. Therefore, an adjustment was made to account for this.

To be able to calculate the adjustments for self-selection suggested by Duffy et al. [13], it is necessary to know the proportion of women attending screening. This is not reported in the paper by Thompson et al. [16] and so the adjusted relative risks cannot be estimated. However, this study is small and therefore omission of this study in the calculation of the pooled relative risk would not have a substantial effect.

The intention to treat estimate,$\;RR_{1}$, was 0.76 (95% CI 0.71–0.83) with significant heterogeneity (*p* < 0.001). This estimate is almost identical to the effect size presented in Figure 2 (RR 0.78). The adjusted pooled relative risk for the effect of being screened, $RR_{2}$, was 0.67 (95% CI 0.61–0.75) and again there was significant heterogeneity between studies with *p* < 0.001 (Figure 3).

When assessing the effect of invitation in women aged 50 and over (Figure 4), whilst the pooled relative risk was similar to that in women of all ages, the *p*-value was 0.175, suggesting no heterogeneity between studies. However, there was still evidence of heterogeneity when assessing the effect of attendance with a relative risk, adjusted for self-selection, of 0.74 (95% CI 0.64–0.85) and a *p*-value of <0.001 (Figure 5).

There were five studies that reported on the effect of screening in women under 50 years. Four studies [18,20,30,53] report the effect of invitation with a pooled relative risk of 0.81 (95% CI 0.74–0.87) and no evidence of heterogeneity (*p* = 0.418). Two studies [15,20] reported the effect of attendance to screening and the adjusted relative risk was 0.73 (95% CI 0.65–0.82).

## 4. Discussion

This systematic review and meta-analysis of IBM studies estimates that the risk of death from breast cancer in women invited for screening is reduced by 22% compared to women not invited (RR 0.78, 95% CI 0.75–0.82), with similar results across age groups. This result is consistent with earlier overviews of cohort studies [5,6] and from the RCTs in breast cancer screening, which suggest invitation to screening reduces mortality by approximately 20% [1].

The studies with contemporaneous control groups are likely to be least biased, with the regional and historical comparison groups potentially affected by differences in the underlying risk of breast cancer mortality between regions or across time periods respectively. When assessing the effect of invitation, the results of studies with contemporaneous control groups ranged from 0.72 (95% CI 0.64–0.79) in the Norwegian study by Weedon-Fekjær [41] to 0.81 (95% CI 0.64–1.01) in the Italian study by Paci [47].

The relative risk, $RR_{2},$ estimates the effect of attendance adjusted for self-selection bias. Using population specific attendance rates and $D_{r}\;$= 1.17 from the SOSSEG study [14] results in a mortality decrease of 33% (RR: 0.67, 95% CI 0.61–0.75). This is slightly more conservative than the relative risk estimated by Broeders et al. [5] although it is unclear whether their result is adjusted for self-selection bias.

The relative risk, $RR_{1}$, estimating the effect of invitation to screening from the relative risk of attendance adjusting for potential self-selection bias, is almost identical to the pooled effect estimated directly. The agreement between the two measures suggests that the adjustment method is valid.

There appears to be significant heterogeneity of the effect of invitation in the meta-analysis of all studies, which disappears when the analysis is stratified by the age. This suggests that there is a difference in the effect of invitation in differing age groups, and that the varying distributions by age among studies is contributing to the heterogeneity of the effect for all ages combined. The heterogeneity of the effect of attendance adjusted for self-selection bias ($RR_{2})$, however, is present for women screened at any age and in women screened over the age of 50 years, suggesting that variation in age distributions is not entirely responsible for heterogeneity among studies. It is likely that this is partly due to differing screening regimens and practices.

Attendance rates varied between studies, from 0.44 in the Canadian study [15] to 0.92 in the study by Sarkeala et al. [32]. Canada differs from other countries in respect of protocol for call and recall, requiring women to self-refer in some provinces. When this opportunistic screening is taken into account, the attendance rate is estimated to be 63.1% [59]. Additionally, the screening programme has a high retention rate, with nearly 80% of previous participants attending a subsequent screen within 36 months.

IBM studies have been the focus of this review but they are not without their limitations. The main limitation is the identification of an appropriate comparison group in the absence of screening [8]. In addition, they are prospective studies and require a long follow-up period to accumulate enough deaths to achieve statistical power [60] and to see the full benefit of screening. Results from the Swedish Two-County Trial suggest that the full benefit of screening requires follow-up of at least 20 years [61]. The majority of studies included in this review had at least 10 years follow-up, with some having over 20 years. In addition, the length of the accrual period should be equal in comparison groups and should be equal to the length of the follow-up period. The effect of screening will be underestimated if the accrual period is shorter than the follow-up period, as more cases will accrue in the screened population than the non-screened population. Seventeen studies in this review had equal accrual and follow-up periods, with ten studies having a shorter accrual period.

Our results update and confirm those of the Euroscreen project for IBM studies [5,8]. Case-control studies reviewed in the Euroscreen project tended to find rather stronger effects than the IBM studies with the effect of invitation 0.69 (95% CI 0.57–0.83) and the effect of attendance adjusted for self-selection bias 0.52 (95% CI 0.42–0.65). This may be due to ascertainment biases in the case-control approach [12]. In addition, the Euroscreen project estimated the effect from trend studies to be between 28–36%, which is comparable to the results in this review [62].

There have been suggestions of alternative analysis methods using the IBM approach. Tabar et al. [63] suggest using as the endpoint the incidence of breast cancers subsequently proving fatal, within ten or twenty years. This method links exposure to endpoint more accurately, but requires a long follow-up period. Sasieni et al. [64] propose a method for estimating the expected number of deaths in the population without screening, which can be used when there is no contemporaneous comparison group. Neither of these methods have been used extensively up to now.

## 5. Conclusions

IBM studies yield estimates uncontaminated by pre-screening cancers. Results from these international studies indicate that inviting women to screening results in a 22% reduction in breast cancer mortality and that the effect of attending screening reduces the risk of death by around 30%. Breast cancer screening in the routine healthcare setting continues to confer a substantial reduction in mortality from breast cancer.

## Figures and Tables

**Figure 1 cancers-12-00976-f001:**
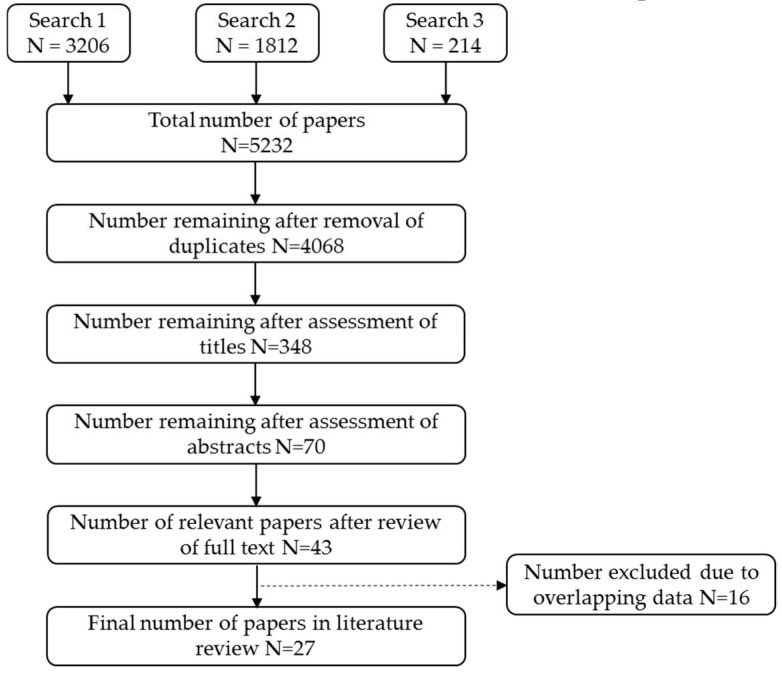
Literature search flow diagram.

**Figure 2 cancers-12-00976-f002:**
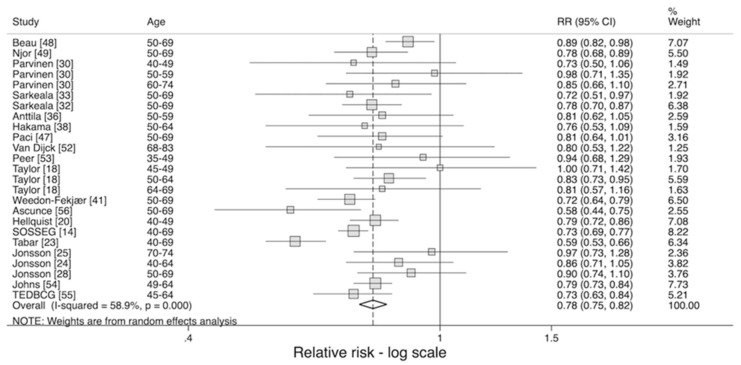
Effect of invitation on risk of breast cancer mortality [14,18,20,23,24,25,28,30,32,33,36,38,41,47,48,49,52,53,54,55,56].

**Figure 3 cancers-12-00976-f003:**
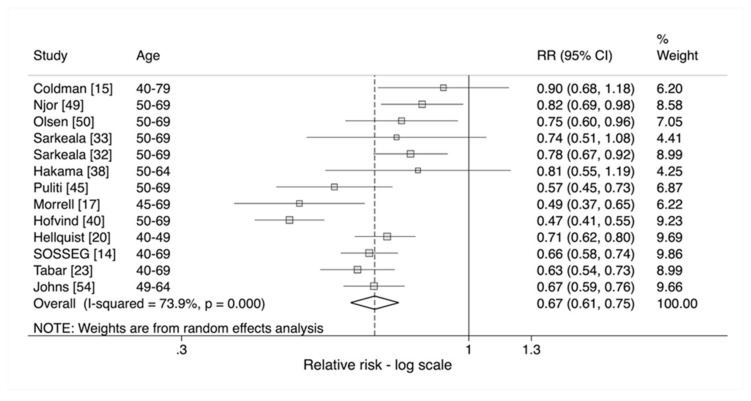
Estimated effect of attendance adjusted for self-selection on risk of breast cancer mortality ($RR_{2}$) [14,15,17,20,23,32,33,38,40,45,49,50,54].

**Figure 4 cancers-12-00976-f004:**
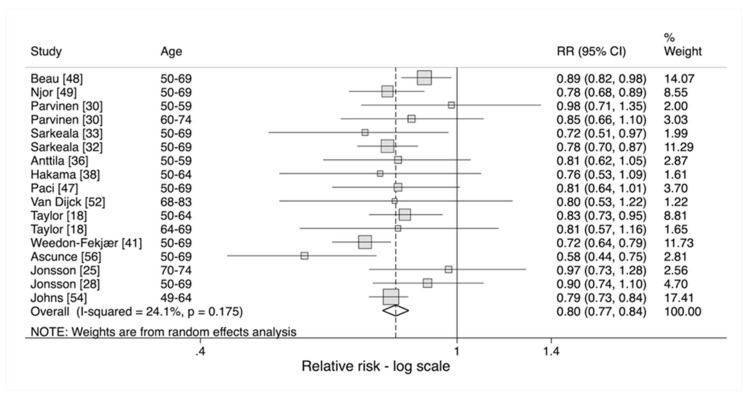
Effect of invitation on risk of breast cancer mortality in women aged 50 and over [18,25,28,30,32,33,36,38,41,47,48,49,52,54,56].

**Figure 5 cancers-12-00976-f005:**
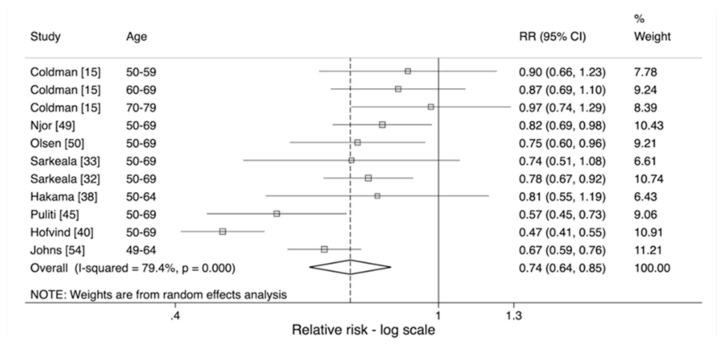
Estimated effect of attendance adjusted for self-selection on risk of breast cancer mortality ($RR_{2}$) in women aged 50 and over [15,32,33,38,40,45,49,50,54].

**Table 1 cancers-12-00976-t001:** Statistical adjustments to estimate effect of invitation and attendance to screening.

$\;RR_{1}:$ Effect of Invitation to Screening	$RR_{2}:$ Effect of Attendance Adjusted for Self-Selection
$\;RR_{1}=D_{r}\left(pRR_{A}+1-p\right)$	$RR_{2}=\frac{pRR_{A}D_{r\;}}{1-\left(1-p\right)D_{r}}$

$RR_{A}$ = the relative risk of breast cancer death associated with attending screening versus not attending; p = the proportion of women who attend screening; $D_{r}$ = the relative risk of breast cancer death for non-attenders versus uninvited = 1.17; Formulae for the variance, and thus the 95% confidence intervals, of these estimates can be found elsewhere [13].

**Table 2 cancers-12-00976-t002:** Overview of Studies.

Reference (by Country and Date)	Region	Age Range of Screening	Screening Interval	Comparison Group(s)	Accrual/follow up Period in Screening Group	Accrual/follow up Period in Comparison Group	Person-Years Study/Comparison Groups(Average Population)
Coldman, 2014 [15]	7 provinces, Canada	40–79 depending on province	2 years ^1^	Contemporaneous	1990–2009	Same	20,155,000
Beau, 2018 [48]	Copenhagen, Denmark	50–69	2 years	Regional and historical	1991–2007	1977–1991 (pre-screening) 1991–2007 (screening)	976,743/17,804,549
Njor, 2015 [49]	Funen, Denmark	50–69	2 years	Regional and historical	1993–2007/09	1979–1993/95 (pre-screening) 1993–2007/09 (screening)	870,465/ 7,096,056; 828,508; 6,151,011
Olsen, 2005 [50]	Copenhagen, Denmark	50–69	2 years	Regional and historical	1991–2001	1981–1991 (pre-screening) 1991–2001 (screening)	430,823/ 634,224; 4,396,417; 4,055,004
Parvinen, 2015 [30]	Turku, Finland	40–74	2 years ^2^	Regional and historical	1987–2009	1976–1986 (pre-screening) 1987–2009 (screening)	853,297/ Helsinki: 2,700,574; Rest of Finland: 21,761,900
Sarkeala, 2008 [33]	8 municipalities, Finland	50–69	2 years	Historical	1992–2003	1974–1985 (pre-screening) 1992–2003 (screening)	228,527
Sarkeala, 2008 [32]	260 municipalities, Finland	50–69	2 years	Historical	1992–2003	1974–1985 (pre-screening) 1992–2003 (screening)	2,731,268
Anttila, 2002 [36]	Helsinki, Finland	50–59	2 years	Contemporaneous ^3^	1986–1997	Same	161,400/155,400
Hakama, 1997 [38]	Finland	50–64	2 years	Contemporaneous	1987–1989/1992	Same	400,804/299,228
Puliti, 2012 [45]	Florence, Italy	50–69	2 years	Contemporaneous	1991–2007/08	Same	50–59: 270,399/113,409 60–69: 233,543/151,615
Paci, 2002 [47]	Florence, Italy	50–69	2 years	Contemporaneous	1990–1996/99	Same	254,890
Van Dijck, 1997 [52]	Nijmegen, Netherlands	68–83	2 years	Regional	1977–1990	1978–1990	60,313/ 61,832
Peer, 1995 [53]	Nijmegen, Netherlands	35–49	2 years	Regional	1975–1990	1976–1990	166,307/ 154,103
Taylor, 2019 [18]	New Zealand	50–64 45–49/ 65–69	2 years	Historical	2001–03/ 2009–11 2006–08/ 2009–11	1996–98/ 2004–06 2001–03/ 2004–06	930,000/766,000 480,000/409,000 249,000/205,000 ^4^
Morrell, 2017 [17]	New Zealand	45–69	2 years	Contemporaneous	1999–2011/ 2000–2011	Same	3,707,483/ 5,405,518
Weedon-Fekjær, 2014 [41]	Norway	50–69	2 years	Contemporaneous	1986–2009	Same	2,407,709/ 12,785,325
Hofvind, 2013 [40]	Norway	50–69	2 years	Contemporaneous ^5^	1996–2009/10	Same	4,814,060/ 988,641
Ascunce, 2007 [56]	Navarre, Spain	50–69	2 years	Historical	1991–2001/ 1997–2001	1980–1990/ 1986–1990	293,000/ 289,000 ^6^
Hellquist, 2011 [20]	Sweden	40–49	1.5–2 years	Regional	1986–2005	Same	7,261,415/ 8,843,852
SOSSEG, 2006 [14]	13 counties, Sweden	40–69 depending on county	2 years	Historical	1980–2001 depending on county	1958–1989 depending on county	7,542,833/ 7,265,841
Tabar, 2003 [23]	Östergötland and Dalarna, Sweden	40–69	1.5–2 years	Historical	1978–1997	1958–1977	2,399,000/ 2,416,000
Jonsson, 2003 [25]	10 counties, Sweden	70–74	2 years	Regional and historical	1986–1998	1976–1988 (pre-screening) 1986–1998 (screening)	1,251,300/ 580,100; 533,400; 1,162,800
Jonsson, 2003 [24]	Gӓvleborg, Sweden	40–64	2 years ^7^	Regional and historical	1974–1986/1998	1964–1973/1985 (pre-screening) 1974–1986/1998 (screening)	855,000/ 2,581,000; 12,619,000
Jonsson, 2001 [28]	7 counties, Sweden	50–69	2 years	Regional and historical	1986–1994/97	1979–1987/1990 (pre-screening) 1987–1993/97 (screening)	2,0360,00/ 1,265,000; 2,046,000; 1,296,000
Johns, 2017 [54]	England and Wales, UK	49–64	3 years	Contemporaneous	1991–2005 ^8^	Same	1,675,356/ 4,719,228
UK Trial of Early Detection of Breast Cancer Group, 1999 [55]	Guildford and Edinburgh, UK	45–64	2 years	Regional	1979–1995	Same	(45,607/ 127,123)
Thompson [16]	Washington, USA	40+ if high risk/50+ if low risk	3 years	Contemporaneous	1982–1988	Same	(94,656)

^1^ Two Provinces, British Columbia and Nova Scotia, invited women aged 40–49 annually; ^2^ Women aged 40–49 were invited yearly for women born in even years, triennially for women born in odd years; ^3^ Invited women born 1935–1939 were compared with uninvited women born in 1930–1934; ^4^ Estimated from data in the paper; ^5^ All women followed from 1986 but screening began in 1995; ^6^ Estimated from data in the paper; ^7^ The average interval between the first and second, and second and third round was 38 months (range 22–65), but was 23 months between rounds 3 and 4; ^8^ The 15-year period 1991–2005 was partitioned into observation periods of two years accrual and up to nine years follow-up.

**Table 3 cancers-12-00976-t003:** Unadjusted and adjusted relative risks.

Reference (by Country and Date)	Country	Age at Screening	Attendance	RR: Unadjusted Effect on Incidence-Based Breast Cancer Mortality		RR Calculated from Effect of Attendance	RR Adjusted for Self-Selection
				Invited Versus not Invited ^1^	Screened Versus not Screened ^1^	$\mathrm{Invited}\;\mathrm{Versus}\;\mathrm{not}\;\mathrm{Invited}\left(RR_{1}\right){\;}^{1}$	$\mathrm{Screened}\;\mathrm{Versus}\;\mathrm{not}\;\mathrm{Screened}\;(RR_{2}){\;}^{1}$
Coldman, 2014 [15]	7 provinces, Canada	40–79 40–49 50–59 60–69 70–79	0.437	NR	0.60 (0.52–0.67) 0.56 (0.45–0.67) ^2^ 0.60 (0.49–0.70) 0.58 (0.50–0.67) 0.65 (0.56–0.74) ^3^	0.97 (0.88–1.06) 0.95 (0.85–1.05) 0.97 (0.87–1.07) 0.96 (0.87–1.05) 0.99 (0.90–1.09)	0.90 (0.68–1.18) 0.84 (0.61–1.16) 0.90 (0.66–1.23) 0.87 (0.69–1.10) 0.97 (0.74–1.29)
Beau, 2018 [48]	Copenhagen, Denmark	50–69	0.71 ^4^	0.89 (0.82–0.98)	NR	NA	NA
Njor,2015 [49]	Funen, Denmark	50–69	0.84	0.78 (0.68–0.89)	0.68 (0.59–0.79)	0.86 (0.75–0.98)	0.82 (0.69–0.98)
Olsen, 2005 [50]	Copenhagen, Denmark	50–69	0.71	0.75 (0.63–0.89) ^5^	0.60 (0.49–0.74)	0.84 (0.72–0.97)	0.75 (0.60–0.96)
Parvinen, 2015 [30]	Turku, Finland	40–49 50–59 60–74	0.867	0.73 (0.50–1.06) 0.98 (0.71–1.35) 0.85 (0.66–1.10)	NR	NA	NA
Sarkeala, 2008 [33]	8 municipalities, Finland	50–69	0.905	0.72 (0.51–0.97)	0.62 (0.43–0.85)	0.77 (0.56–1.06)	0.74 (0.51–1.08)
Sarkeala, 2008 [32]	260 municipalities, Finland	50–69	0.924	0.78 (0.70–0.87)	0.66 (0.58–0.75)	0.80 (0.70–0.92)	0.78 (0.67–0.92)
Anttila, 2002 [36]	Helsinki, Finland	50–59	0.82	0.81 (0.62–1.05)	NR	NA	NA
Hakama, 1997 [38]	Finland	50–64	0.85	0.76 (0.53–1.09)	0.67 (0.46–0.97) ^6^	0.84 (0.62–1.15)	0.81 (0.55–1.19)
Puliti, 2012 [45]	Florence, Italy	50–69 50–59 60–69	0.64	NR	0.44 (0.36–0.54) ^7^ 0.55 (0.41–0.75) 0.49 (0.38–0.64)	0.75 (0.67–0.85) 0.83 (0.71–0.98) 0.79 (0.68–0.91)	0.57 (0.45–0.73) 0.71 (0.51–0.98) 0.63 (0.47–0.85)
Paci, 2002 [47]	Florence, Italy	50–69	0.60	0.81 (0.64–1.01)	NR	0.75 (0.67–0.85)	0.57 (0.45–0.73)
Van Dijck, 1997 [52]	Nijmegen, Netherlands	68–83	0.46	0.80 (0.53–1.22)	NR	NA	NA
Peer,1995 [53]	Nijmegen, Netherlands	35–49	0.65	0.94 (0.68–1.29)	NR	NA	NA
Taylor, 2019 [18]	New Zealand	45–49 50–64 65–69	0.72	1.00 (0.71–1.42) ^8^ 0.83 (0.73–0.95) 0.81 (0.57–1.16)	NR	NA	NA
Morrell, 2017 [17]	New Zealand	45–69	0.64	NR	0.38 (0.30–0.49)	0.71 (0.62–0.80)	0.49 (0.37–0.65)
Weedon-Fekjær, 2014 [41]	Norway	50–69	0.76	0.72 (0.64–0.79)	NR	NA	NA
Hofvind, 2013 [40]	Norway	50–69	0.84	NR	0.39 (0.35–0.44)	0.57 (0.51–0.64)	0.47 (0.41–0.55)
Ascunce, 2007 [56]	Navarre, Spain	50–69	0.85	0.58 (0.44–0.75) ^9^	NR	NA	NA
Hellquist, 2011 [20]	Sweden	40–49	0.80	0.79 (0.72–0.86)	NR	0.74 (0.66–0.83) ^10^	0.71 (0.62–0.80)
SOSSEG, 2006 [14]	13 counties, Sweden	40–69	0.80 ^11^	0.73 (0.69–0.77)	0.55 (0.51–0.59)	0.75 (0.68–0.82)	0.66 (0.58–0.74)
Tabar, 2003 [23]	Östergötland and Dalarna, Sweden	40–69	0.85	0.59 (0.53–0.66)	0.52 (0.46–0.59)	0.69 (0.61–0.78)	0.63 (0.54–0.73)
Jonsson, 2003 [25]	10 counties, Sweden	70–74	0.84 ^12^	0.97 (0.73–1.28)	NR	NA	NA
Jonsson, 2003 [24]	Gӓvleborg, Sweden	40–64	0.84	0.86 (0.71–1.05)	NR	NA	NA
Jonsson, 2001 [28]	7 counties, Sweden	50–69	0.80 ^11^	0.90 (0.74–1.10)	NR	NA	NA
Johns, 2017 [54]	England and Wales, UK	49–64	0.74	0.79 (0.73–0.84)	0.54 (0.51–0.57)	0.77 (0.71–0.84)	0.67 (0.59–0.76)
UK Trial of Early Detection of Breast Cancer Group, 1999 [55]	Guildford and Edinburgh, UK	45–64	0.654	0.73 (0.63–0.84)	NR	NA	NA
Thompson, 1994 [16]	Washington, USA	≥50	NR	NR	0.61 (0.23–1.62)	NA	NA
Pooled RR from all studies		Random effects Heterogeneity *p*-value		0.78 (0.75–0.82) <0.001	0.54 (0.49–0.59) <0.001	0.76 (0.71–0.83) <0.001	0.67 (0.61–0.75) <0.001
Pooled RR from studies inviting women aged 50 and over		Random effects Heterogeneity *p*-value		0.80 (0.77–0.84) 0.175	0.57 (0.51–0.64) <0.001	0.82 (0.74–0.92) <0.001	0.74 (0.64–0.85) <0.001
Pooled RR from studies inviting women under 50 years		Random effects Heterogeneity *p*-value		0.81 (0.74–0.87) 0.418	0.56 (0.45–0.67) ^13^–	0.84 (0.66–1.06) 0.002	0.73 (0.65–0.82) 0.343

^1^ RR, relative risk (95% confidence interval), NR, not reported, NA, not applicable; ^2^ Pooled RR of three provinces; ^3^ Pooled RR of four provinces; ^4^ Participation rate not reported so taken from Olsen AH, Njor SH, Vejborg I, Schwartz W, Dalgaard P, Jensen, M.B.; et al. Breast cancer mortality in Copenhagen after introduction of mammography screening: cohort study. *BMJ*
**2005**, *330*, 220. [50]; ^5^ Not included in meta-analysis due to later paper by Beau et al.; ^6^ Calculated from data in the paper; ^7^ Unadjusted RR calculated from data in the paper; ^8^ RR calculated from data in the paper. The accrual period for women aged 50–64 was 2001–2003 and for women aged 45–49 and 65–69 was 2006–2008; ^9^ Excludes prevalent cases; ^10^ RR1 and RR2 are as reported by authors in the paper; ^11^ Attendance rate not reported so taken from Giordano L, von Karsa L, Tomatis M, Majek O, de Wolf C, Lancucki L, et al. Mammographic screening programmes in Europe: organization, coverage and participation. *J. Med Screen*. **2012**, *19*, 72–82. [57] as region reported invites women until the age of 74; ^12^ Attendance rate not reported so taken from Swedish Organised Service Screening Evaluation Group. Reduction in breast cancer mortality from the organised service screening with mammography: 2. Validation with alternative analytic methods. *Cancer Epidemiol Biomark. Prev*. **2006**, *15*, 52–56. [58]; ^13^ One study only.

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
