# Peer review of "Worldwide Review and Meta-Analysis of Cohort Studies Measuring the Effect of Mammography Screening Programmes on Incidence-Based Breast Cancer Mortality"

_cancers, 2020, doi:10.3390/cancers12040976_

Round 1

Reviewer 1 Report

This is an interesting overview. 

the authors claim that screening has substantial reduction in mortality from breast cancer, and cite a hazard ratio of 0.78

for invitation to screening versus unscreened. 

the hazard ratio is actually for a subgroup of the population who is eligible for screening.  consider the deaths from the population for women diagnosed under 40 (or 50)  and for those diagnosed over 70 (or 75). this is about one third of the deaths.     Then consider the uptake of screening in the population.  The actual crude reduction in mortality in the entire population is closer to ten percent. HR= 0.9   It is not valid to extrapolate the benefit of screening to all women in the population unless all are candidates for screening.  With this in mind it is not obvious to promote screening and it is nonsense to  communicate the message as early detection is the key to breast cancer.

Also there is a unique bias which is not considered.  if we compare women who are ever screened and never screened those who are never screened are more likely to get breast cancer.  Once a woman gets breast cancer she is not eligible for screening.  The longer a woman is followed cancer free, the more time she is eligible to get screened. So screening  per se is associated with survival.   We have modeled this and get a pretty strong bias in favor of screening.   Is this bias a consideration here.    

Bias is of course of major concern.   Would be helpful to list the possible biases that are at play here and discuss why they are not pertinent.

Having studied this closely for years I am more and  more convinced that the Canadian trial  got it right. 

Author Response

Reviewer 1:

This is an interesting overview.

the authors claim that screening has substantial reduction in mortality from breast cancer, and cite a hazard ratio of 0.78 for invitation to screening versus unscreened. the hazard ratio is actually for a subgroup of the population who is eligible for screening.  consider the deaths from the population for women diagnosed under 40 (or 50)  and for those diagnosed over 70 (or 75). this is about one third of the deaths.     Then consider the uptake of screening in the population.  The actual crude reduction in mortality in the entire population is closer to ten percent. HR= 0.9   It is not valid to extrapolate the benefit of screening to all women in the population unless all are candidates for screening.  With this in mind it is not obvious to promote screening and it is nonsense to  communicate the message as early detection is the key to breast cancer.

Author response: We now clarify in the methods that the reductions in mortality with invitation pertain to those populations actually offered screening and the reductions associated with screening itself pertain to the populations taking up the offer. See page 2, lines 65 – 68.

Also there is a unique bias which is not considered.  if we compare women who are ever screened and never screened those who are never screened are more likely to get breast cancer.  Once a woman gets breast cancer she is not eligible for screening.  The longer a woman is followed cancer free, the more time she is eligible to get screened. So screening per se is associated with survival.   We have modeled this and get a pretty strong bias in favor of screening.  Is this bias a consideration here.   

Author response: In fact, women who choose to participate in screening are at higher risk of breast cancer than those who do not. See for example Beckmann et al, J Med Screen 2013; 20: 208-19). There is evidence that despite this, women who do not choose to be screened may have higher mortality from breast cancer due to poorer survival, hence our correction for self-selection bias. See page 2, section 2.3.1 Adjustment for Self-selection bias. The reviewer’s concern that there may be an element of reverse causality due to a longer time without cancer meaning a longer time to be screened is avoided in the studies reviewed here either by design, for estimation of the effect of invitation, or in analysis for the effect of being screened, since this is defined on the basis of attendance following most recent invitation.

Bias is of course of major concern.   Would be helpful to list the possible biases that are at play here and discuss why they are not pertinent.

Author response: The main risk of bias comes from self-selection, hence our correction for this. Again, see page 2, section 2.3.1 Adjustment for Self-selection bias.

Having studied this closely for years I am more and more convinced that the Canadian trial  got it right

Author response: The reviewer is entitled to his or her opinion. However, it has been observed by others that there are serious risks of bias in the Canadian trials (see for example Heywang-Köbrunner et al, Eur Radiol 2016; 26: 342-50, and Boyd et al, Radiol 1993; 198: 661-3). We do not discuss these as it is not the purpose of our paper to describe the shortcomings of the Canadian trials. Our purpose is to summarise results from incidence-based mortality studies of screening as it is practised now.

Reviewer 2 Report

This paper presents an extension of the Euroscreen project, from 2012 to October 2019, and from Europe to the world (noting that for the latter, the authors only include data from Canada, USA, and New Zealand, thus leaving out all of South America, Africa and Asia). The authors’ meta-analysis looks at mortality (and not other epidemiological data such as cancer rates per race, etc), which is clearly described in the manuscript. They look at the effects of both invitation to screening and actual attendance at screening. I agree with the authors’ conclusion that the routine screening helps reduce breast cancer mortality (and not just breast cancer incidence). I find it interesting that the authors looked separately at the invitation to screening and attendance at screening. I would hope that these data will help epidemiologist gain more information on attendance from different populations subsets (by race, socio-economic background, education level, rural vs. urban, distance to closest clinic, etc).

This is a well-written and well-presented review that merits publication in Cancers. I find it particularly timely given all of the recent (past 10-15 years or so) literature about breast cancer overdiagnoses and unnecessary procedures. However, such topics are not discussed within.

The introduction is concise, but well-articulated. While the authors mention that the age of study participants was not restricted, I wasn’t able to find out 1) why they didn’t decide to include this parameter, nor 2) whether they plan to do so in the future.

It didn’t seem clear to me why the authors chose age 50 as a threshold to segregate the age groups.

The data tables are well presented and adequately labeled.

Line 165: Spaces should be inserted before and after the heading for Section 3.3.

Figures 2, 3, 4, 5 have a font size that is too small and they are a bit blurred.

Author Response

Reviewer 2:

This paper presents an extension of the Euroscreen project, from 2012 to October 2019, and from Europe to the world (noting that for the latter, the authors only include data from Canada, USA, and New Zealand, thus leaving out all of South America, Africa and Asia). The authors’ meta-analysis looks at mortality (and not other epidemiological data such as cancer rates per race, etc), which is clearly described in the manuscript. They look at the effects of both invitation to screening and actual attendance at screening. I agree with the authors’ conclusion that the routine screening helps reduce breast cancer mortality (and not just breast cancer incidence). I find it interesting that the authors looked separately at the invitation to screening and attendance at screening. I would hope that these data will help epidemiologist gain more information on attendance from different populations subsets (by race, socio-economic background, education level, rural vs. urban, distance to closest clinic, etc).

This is a well-written and well-presented review that merits publication in Cancers. I find it particularly timely given all of the recent (past 10-15 years or so) literature about breast cancer overdiagnoses and unnecessary procedures. However, such topics are not discussed within.

Author response: We thank the reviewer for the remarks. It is true that our paper focusses only on breast cancer mortality. We are working on reviews and primary studies of the other issues, but we feel the paper is broad enough as it is.

The introduction is concise, but well-articulated. While the authors mention that the age of study participants was not restricted, I wasn’t able to find out 1) why they didn’t decide to include this parameter, nor 2) whether they plan to do so in the future.

Author response: In general the age ranges were chosen for us by the studies reviewed. The majority dealt with ages 50-69, and a relatively small number consider screening in the 40’s and 70’s. We now state this explicitly. See page 2, lines 52 and 69 – 73.

It didn’t seem clear to me why the authors chose age 50 as a threshold to segregate the age groups.

Author response: We chose this age as so much debate over the years has focussed on whether it is effective or cost-effective to screen below this age threshold. We now state this. See page 2, lines 69 – 73.

The data tables are well presented and adequately labeled.

Line 165: Spaces should be inserted before and after the heading for Section 3.3.

Author response: Done. Now line 173.

Figures 2, 3, 4, 5 have a font size that is too small and they are a bit blurred.

Author response: Done. Figures amended. See pages 13 – 16.

Reviewer 3 Report

In this study, Dibden et al. performed meta-analysis based on published literature and investigated the effect of mammography screening invitation on incidence-based breast cancer mortality. The manuscript was logical and well written. My major concern is the novelty of the research approach and findings. A previous study published in Journal of Medical Screening in 2012 asked the same question and presented similar data in multiple European countries. Compared to that publication, this manuscript incorporated new data since 2012, and included USA, Canada, and New Zealand as well. However, this expansion in scope was not substantial, and the findings do not offer novel significant insights. Study itself is solid though.

Author Response

Reviewer 3:

In this study, Dibden et al. performed meta-analysis based on published literature and investigated the effect of mammography screening invitation on incidence-based breast cancer mortality. The manuscript was logical and well written. My major concern is the novelty of the research approach and findings. A previous study published in Journal of Medical Screening in 2012 asked the same question and presented similar data in multiple European countries. Compared to that publication, this manuscript incorporated new data since 2012, and included USA, Canada, and New Zealand as well. However, this expansion in scope was not substantial, and the findings do not offer novel significant insights. Study itself is solid though.

Author response: We submit that the fact that updating and broadening the scope to worldwide does not greatly change the conclusions is of itself interesting news. It adds to the evidence that early detection continues to prevent breast cancer deaths in our epoch of effective systemic therapies.

Round 2

Reviewer 3 Report

Little has changed from the original version; hence
my view on the impact of the paper remains the same. In terms of the
technicality of the study, I had no major complaints.